# The evolution and structure of snake venom phosphodiesterase (svPDE) highlight its importance in venom actions

Cheng-Tsung Pan[1,2], Chien-Chu Lin[1†], I-Jin Lin[1], Kun-Yi Chien[3], Yeong-Shin Lin[4]*, Hsiao-Han Chang[1,5,6]*, Wen-Guey Wu[1,5]*

[1]Institute of Bioinformatics and Structural Biology, National Tsing Hua University, Hsinchu, Taiwan; [2]Department of Biostatistics, University of Oslo, Oslo, Norway; [3]Graduate Institute of Biomedical Sciences, Department of Biochemistry and Molecular Biology, College of Medicine, Chang Gung University, Taoyuan, Taiwan; [4]Institute of Bioinformatics and Systems Biology, National Yang Ming Chiao Tung University, Hsinchu, Taiwan; [5]Department of Life Science, National Tsing Hua University, Hsinchu, Taiwan; [6]Institute of Molecular and Cellular Biology, National Tsing Hua University, Hsinchu, Taiwan

**\*For correspondence:**
yslin@nycu.edu.tw (Y-SL);
hhchang@life.nthu.edu.tw (H-HC);
wgwu@life.nthu.edu.tw (W-GW)

**Present address:** [†]Division of Chemical Biology and Medicinal Chemistry, Eshelman School of Pharmacy, University of North Carolina at Chapel Hill, Chapel Hill, United States

**Competing interest:** The authors declare that no competing interests exist.

**Abstract** For decades, studies of snake venoms focused on the venom-ome-specific toxins (VSTs). VSTs are dominant soluble proteins believed to contribute to the main venomous effects and emerged into gene clusters for fast adaptation and diversification of snake venoms. However, the conserved minor venom components, such as snake venom phosphodiesterase (svPDE), remain largely unexplored. Here, we focus on svPDE by genomic and transcriptomic analysis across snake clades and demonstrate that soluble svPDE is co-opted from the ancestral membrane-attached ENPP3 (ectonucleotide pyrophosphatase/phosphodiesterase 3) gene by replacing the original 5′ exon with the exon encoding a signal peptide. Notably, the exons, promoters, and transcription/translation starts have been replaced multiple times during snake evolution, suggesting the evolutionary necessity of svPDE. The structural and biochemical analyses also show that svPDE shares the similar functions with ENPP family, suggesting its perturbation to the purinergic signaling and insulin transduction in venomous effects.

## Editor's evaluation

This manuscript reports important findings regarding the evolution of snake venom proteins. The conclusions are convincing and are based on appropriate and validated methodologies in line with the current state-of-the-art. The findings will be of interest to biologists and biochemists interested in the evolution of venoms as well as those generally interested in the evolution of molecular novelties.

## Introduction

Snakebite envenoming is a neglected tropical disease that leads to over 100,000 deaths worldwide annually (*Gutiérrez et al., 2017*), and its mortality may be even higher than malaria in some regions (*Stock et al., 2007*). For decades, scientists have been dedicated to discovering the components of snake venoms. The diversity and species specificity of snake venoms make the first aid and treatments difficult for snakebites. Traditionally, the dominant proteins in the venoms have been identified by proteomic approaches as the major components of venoms. Recent advances in whole-genome

sequencing of venomous snakes have prompted many studies to consider the origin and diversification of such dominant proteins, which have been named as the venom-ome-specific toxins (VSTs) (*Suryamohan et al., 2020*) and have been regarded as the main contributors of toxic effects.

Most VSTs have evolved from ancestral genes and been duplicated into large gene clusters in several snake clades. For example, the gene clusters of snake venom metalloproteases (SVMP) in Viperidae and three-finger toxins (3FTx) in Elapidae have been revealed to evolve from the ancestral genes of a membrane-anchored protein (named ADAM28 in other species) and a membrane GPI-anchored protein (named Ly6 in other species), respectively (*Almeida et al., 2021*; *Giorgianni et al., 2020*; *Margres et al., 2021*; *Rao et al., 2022*). Alternative mRNA splicing at the 3′ end has been proposed to account for the adaptation of encoding ancestral membrane-anchored ADAM28 proteins into a soluble protein in Viperidae by losing the C-terminal transmembrane domain. This ancestral gene has evolved into the SVMP gene cluster with venom-specific expression in Viperidae followed by multiple gene duplication events (*Giorgianni et al., 2020*; *Ogawa et al., 2019*). Another example is the 3FTx in Elapidae, which may have evolved from the gene of GPI-anchored Ly6 protein by losing the sequence encoding the peptide responsible for the attachment to GPI lipids on the membrane (*Tsetlin, 2015*).

Aside from the VSTs mentioned above, the low quantity of minor venom components hinders researchers from studying them, rendering their biological significance elusive. One minor venom component, snake venom phosphodiesterase (svPDE), has been found to be ubiquitous in the venoms of most venomous species in Viperidae (~0.01–2.5%) (*Damm et al., 2021*) and Elapidae (~0.4–1.1%) (*Laustsen et al., 2015*; *Tan et al., 2015*). SvPDE is a soluble high-molecular-weight glycoprotein and is distinct from the intracellular phosphodiesterase (*Al Saleh and Khan, 2011*; *Mitra and Bhattacharyya, 2014*; *Oliveira et al., 2021*; *Peng et al., 2011*; *Santoro et al., 2009*; *Trummal et al., 2014*; *Valério et al., 2002*). Since extracellular ATP is involved in epithelial homeostasis (*Mori et al., 2022*) and also functions as a danger signal of damaged cells through the purinergic signaling pathway (*Burnstock, 2016*; *Cintra-Francischinelli et al., 2010*), svPDE, which enzymatically acts on the extracellular ATP, is expected to perturb the related physiological responses. However, the gene encoding svPDE, the evolutionary origin of that gene, and the potential biological targets of svPDE remain largely unknown. In this study, we integrated comparative genomics, comparative transcriptomics of the venom glands, and the biochemical analysis of svPDE obtained directly from the venom to investigate the genomic locus, gene structure, evolution, protein structure, enzymatic activities, and potential targets of svPDE from *Naja atra*.

## Results

### The evolution of the soluble secretory svPDE from the membrane-anchored ENPP3 via alternative splicing of their encoding genes

The presence of svPDE in venoms has been identified by proteomics or by detecting its enzymatic activity for decades. However, its gene characteristics have not been well reported. To reveal the genomic locus and the gene structure of svPDE, we integrated genomic, transcriptomic and proteomic data. The query peptide sequence of svPDE was retrieved from PDB database (accession: 5GZ4). This svPDE was directly purified from the crude venom of *N. atra* captured in Taiwan. The target genomes included the draft one of *N. atra* sequenced from a muscle tissue (ongoing internal project, see 'Materials and methods') and the complete one of its sister species, *Naja naja*, from the public data (*Suryamohan et al., 2020*). For genomes of *N. atra* and *N. naja*, the only genomic hit of the svPDE peptide sequence is located on the ENPP3 locus inferred from the conserved synteny, while the peptide sequence is a subsequence of the translated sequence inferred from the putative exons of the ENPP3 gene (*Figure 1—figure supplement 1*). Consistently, compared with the ENPP3 peptide sequence, the purified svPDE is shorter at the N-terminal, lacking the cytoplasmic (CP) and transmembrane (TM) domains of ENPP3 (*Figure 1—figure supplement 1*). The unique genomic hit, conserved synteny, and identical amino acid sequences strongly suggest that svPDE and ENPP3 are encoded from the same genomic locus and share all but the 5′ end exons.

The absence of CP and TM domains in purified svPDE protein is consistent with the secretory feature of most venom proteins (*Figure 1A* and *Figure 1—figure supplement 1*). However, as the ENPP3 gene is highly conserved in the entire Metazoan clades (*Figure 1—figure supplement 2*)

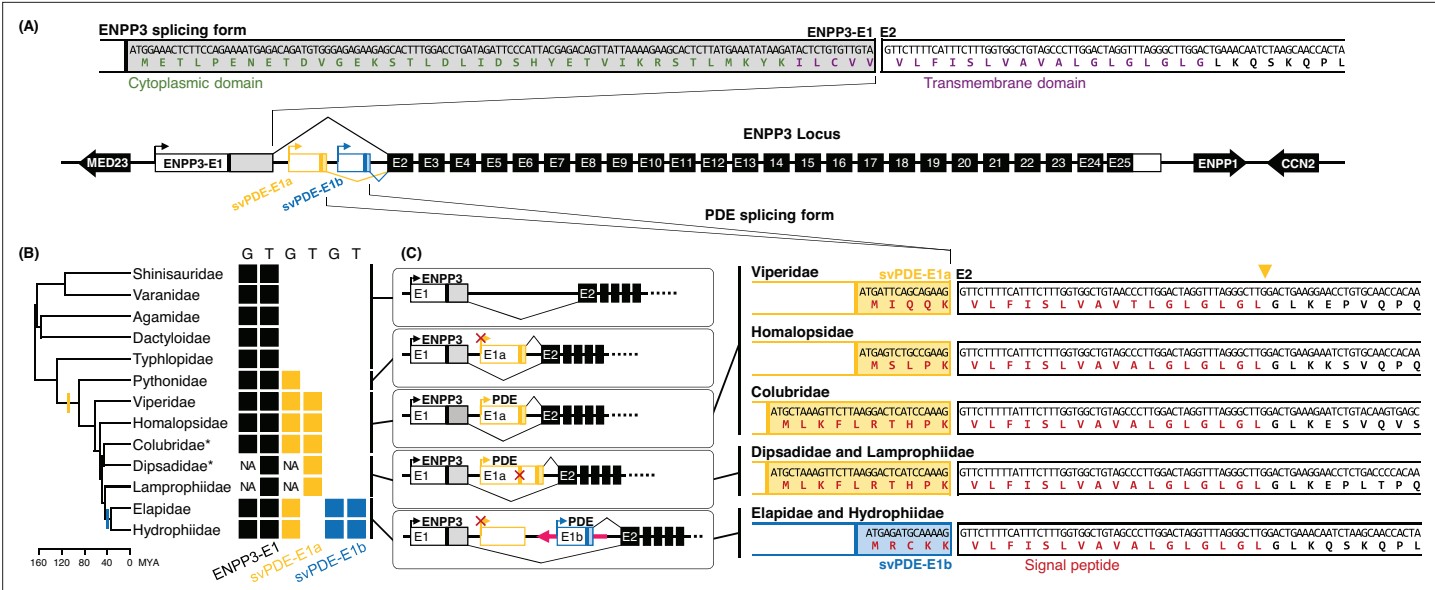

**Figure 1.** The alternative splicing and evolutionary features of ENPP3 and svPDE. (**A**) N-terminal of peptide sequences of ENPP3 and PDE encoded by alternative splicing. The scheme of exons and genomic synteny is presented with flanking coding genes. The 5' untranslated regions (5' UTR) and coding regions of first exons are separated by colored vertical lines, which are the translation start codons. Only the lengths of E1, E1a, and E1b are drawn in scale. Putative promoters identified in the proximal upstream are indicated with arrows attached to the first exons. The same elements are expressed by the same colors throughout the figure. The sequences around splicing junctions of E1 and E2 are zoomed in with translated peptides of functional domains highlighted in corresponding colors. (**B**) Presences and absences of ENPP3-specific and svPDE-specific E1s in different clades. Color-filled squares denote the exons identified in the genomes 'G' and transcriptomes 'T,' where 'NA' indicates the clades without available genome assemblies. Clades with asterisks use Duvernoy's glands as their venom delivery system. The colored vertical lines on the phylogenetic tree indicate the inferred branches that svPDE-E1a (yellow) and svPDE-E1b (blue) emerged. (**C**) The scheme of alternative splicing of ENPP3 and svPDE transcripts in different clades. The possible malfunctioned elements, including svPDE-E1a promoters in Pythonidae, Elapidae, and Hydrophiidae, and ancestral translation start sites in Dipsadidae and Lamprophiidae, are crossed out in red. The long pink arrow indicates the mobile element. The yellow triangle indicates the putative cleavage site of the signal peptidase.

The online version of this article includes the following figure supplement(s) for figure 1:

**Figure supplement 1.** The sequence alignment of the translated peptides of ENPP3 inferred from the genomic exons (*N. naja* and *N. atra*) and the snake venom phosphodiesterase (svPDE) peptides revealed from proteomic approaches.

**Figure supplement 2.** Presences and absences of ENPP family members across eukaryotes.

**Figure supplement 3.** The multiple sequence alignment of the translated peptides around the E1-E2 junction region.

**Figure supplement 4.** The multiple sequence alignment of (**A**) the putative core promoters located on the upstream of svPDE-E1a, (**B**) the alternative translation start sites of svPDE-E1a in Dipsadidae and Lamprophiidae, and (**C**) the putative core promoters located on the upstream of svPDE-E1b.

**Figure supplement 5.** The multiple sequence alignment of the coding regions of svPDE-E1a and 5' partial E2.

and is essential for the basic biochemical functions (*Borza et al., 2022*), instead of replacing original ENPP3 by svPDE, their transcripts are expected to coexist in snakes. One possibility for their coexistence is through an alternative splicing mechanism (*Sorek, 2007*; *Verta and Jacobs, 2022*), in which alternative 5' exons with a signal peptide emerged, leading to the soluble secretory form without damaging the functions of the remaining peptides. To test our hypothesis, we comprehensively de novo assembled transcriptomes from the species across 13 clades of Toxicofera (*Figure 1B*) with publicly available RNA-seq data and compared them with the corresponding genomes available in the NCBI Assembly database (see *Supplementary file 1* for sample details). In addition to the transcript encoding the conserved ENPP3 protein, we found potential svPDE transcripts comprising all but the first (E1) exons of the ENPP3 gene.

A careful comparison reveals two types of novel clade-specific E1 of svPDE transcripts (*Figure 1B* and *Figure 1—figure supplement 3*). The first one, named svPDE-E1a, found in the transcriptomes of all snake clades but Elapidae and Hydrophiidae; and the other, named svPDE-E1b, found only in Elapidae and Hydrophiidae. Both svPDE-E1a and svPDE-E1b accompany a putative TATA-box core promoter located on their proximal upstream (*Figure 1A* and *Figure 1—figure supplement 4*) and

encode signal peptides necessary for the secretory feature of svPDE (*Figure 1C* and *Figure 1—figure supplement 3*). In addition, the canonical GT-AG introns between the novel E1 (for both two types) and the conserved E2 were consistently observed in all snake species with available genomes. Together, our results suggest that the svPDE protein came from the co-option of the ancestral ENPP3 gene by using a novel 5′ exon.

## The recruitment of svPDE

We next traced the origin of the novel E1 in Toxicofera by examining transcriptomic and genomic data. We found that the genomic sequences of svPDE-E1a were present and conserved in all species of Serpentes except for the earliest diverged Typhlopidae. This suggests an early emergence of svPDE-E1a in the Serpentes evolution and became nonfunctional in some descent lineages (i.e., not expressed in Pythonidae, Elapidae, and Hydrophiidae) (*Figure 1B*). Pythonidae has genomic mutations in the TATA-box of svPDE-E1a (*Figure 1C* and *Figure 1—figure supplement 4A*), and the absence of an alternative mechanism to express svPDE is probably associated with its non-venomous character (*Reyes-Velasco et al., 2015*). For svPDE-E1a in some species of Dipsadidae and Lamprophiidae, although their original translation start codons (ATG) had mutated, alternative ATGs were recruited at the downstream region of the original ones (*Figure 1C* and *Figure 1—figure supplement 4B*). This recruitment of start codons was caused by a newly 3′ splicing site of svPDE-E1a emerged in their common ancestor (*Figure 1—figure supplement 5*) that elongates the length of svPDE-E1a at the 3′ end. This replacement allows for the possibility of svPDE translation in Dipsadidae and Lamprophiidae.

Similar replacements of elements involved in svPDE expression were also found in Elapidae and Hydrophiidae: the emergence of exon svPDE-E1b. Although svPDE-E1a is highly conserved in the genomes of Elapidae and Hydrophiidae, it was not found in their transcriptomes, potentially due to the genomic mutation of a TATA-box at the putative core promoter for svPDE-E1a in their common ancestor (*Figure 1C* and *Figure 1—figure supplement 4A*). Instead, their svPDE transcripts include another 5′ exon, svPDE-E1b, which was not identified in the genomes of other clades that diverged earlier. These results indicate that svPDE transcripts switched from using svPDE-E1a to svPDE-E1b in the common ancestor of Elapidae and Hydrophiidae (*Figure 1C*).

Notably, the entire svPDE-E1b, including its putative promoter, is embedded in the antisense strand of an uncharacterized Elapidae mobile element (Dfam accession: DR0148352, *Figure 1—figure supplement 4C*) that inserted into the ENPP3 locus in the common ancestor of Elapidae and Hydrophiidae (*Figure 1C*). We examined other DR0148352 elements in the genome-available species of Elapidae and Hydrophiidae and found that the homologous promoter region in the elements do not contain TATA-box patterns or other promoter-related domains (*Figure 1—figure supplement 4C*). This suggests that the putative promoter of svPDE-E1b emerged in situ after the mobile element insertion. Interestingly, the efficiency of svPDE expression could be associated with the TATA-box of svPDE-1b since Elapidae and Hydrophiidae express significantly high and low levels of svPDE (*Figure 2A*) in line with the canonical and non-canonical TATA-box patterns (*Figure 1—figure supplement 4C*), respectively.

## SvPDE are predominantly expressed in the venom glands of venomous snakes

We then compared the svPDE expression in venom glands among different snake clades using transcriptomic data. As the sequences of svPDE and ENPP3 only differ in the first exon, the reads mapped to the shared exons cannot be distinguished from different transcripts. Thus, we estimated the relative expression of svPDE over ENPP3 by the difference in the number of reads spanning distinct E1-E2 junctions of ENPP3 and svPDE splicing forms. This method also reduced the potential biases caused by different experiments, independent samples, the uncertainty of 5′ UTR length due to library preparation, and the uneven distribution of mapped reads.

As shown in *Figure 2A*, for the relative differences of svPDE over ENPP3 expression, most species of Viperidae, Lamprophiidae, and Elapidae exhibit a predominance of svPDE in their venom glands, in line with the current understanding of venoms by proteomic studies in such clades (*Damm et al., 2018*; *Gren et al., 2019*). The relative expression of svPDE was inferior to ENPP3 in the glands of Homalopsidae, Colubridae, Dipsadidae, and Hydrophiidae (*Figure 2A*). Intriguingly, compared to the deficient expression in the other three clades, the expression of svPDE in Homalopsidae was higher,

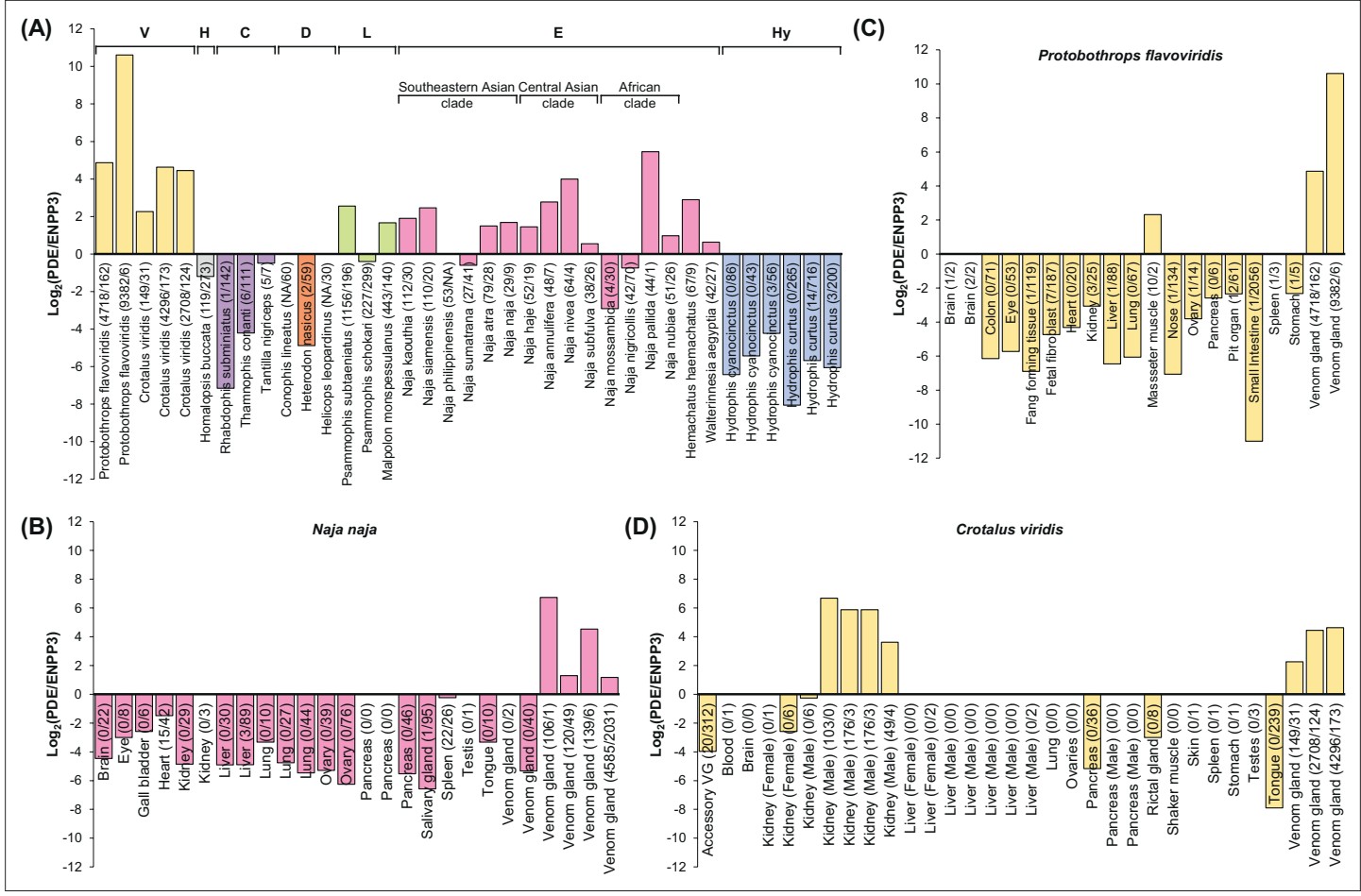

**Figure 2.** Species-specific and tissue-specific expression of snake venom phosphodiesterase (svPDE) over ENPP3 splicing forms. The expression of each splicing form was calculated by the number of reads spanning the specific E1s and the conserved E2. The log2 transformed ratios are shown in the figure and a count of zero was adjusted to a pseudo count of one for valid log transformation. The unadjusted counts are presented in the axis labels as the format of svPDE/ENPP3. The alternative splicing forms not identified in the transcriptomes and no available genomes to infer their presence are denoted as NA. (**A**) Species-specific expressions in venom glands and, for Colubridae and Dipsadidae, Duvernoy's gland, an anatomic gland structure similar to venom glands, were compared. Samples with less than five svPDE transcripts and less than five ENPP3 transcripts are regarded as having no expression for both transcripts and not shown. Clades are colored in yellow, gray, purple, orange, green, pink, and blue with the clade prefixes of Viperidae, Homalopsidae, Colubridae, Dipsadidae, Lamprophiidae, Elapidae, and Hydrophiidae, respectively. (**B–D**) Tissue-specific expression of svPDE over ENPP3 in selected species: (**B**) *Naja naja* (Elapidae) and (**C**) *Protobothrops flavoviridis* (Viperidae) and (**D**) *Crotalus viridis* (Viperidae).

although not larger than ENPP3. This feature provides new insight into the characteristics of Homalopsidae, a species still being discovered (*Bernstein et al., 2021*; *Köhler et al., 2021*). In contrast to the predominance of svPDE across the species of Viperidae, the svPDE expression across the species of Elapidae shows higher interspecies variation and even exhibits a contrary pattern in some of the species (*Figure 2A*). Proteomic studies have identified this variation in Elapidae for decades (*Modahl et al., 2020*; *Tan and Tan, 1988*), but it still requires future functional studies of svPDE in the venoms to understand the significance of this variation.

## The tissue-specific expression of svPDE

Traditionally, svPDE studies have only focused on the venom gland by detecting its enzyme activity and neglected the expression in other tissues. We identified the difference between svPDE and ENPP3 transcripts. The reads originated from two transcripts that became distinguishable in the transcriptomic data. We investigated the relative expression of svPDE over ENPP3 in other tissues to understand their tissue-specificity and whether they are mutually exclusive. In both Elapidae (*Figure 2B*) and Viperidae (*Figure 2C and D*), taking India cobra (*N. naja*), habu (*Protobothrops flavoviridis*), and

prairie rattlesnake (*Crotalus viridis*) as examples, venom glands exhibit excessively high svPDE levels and very low ENPP3 expression. On the other hand, ENPP3 is primarily abundant in small intestines, and for those tissues enriched with ENPP3, the frequency of svPDE transcripts is low or nonexistent. These results indicate the mutually exclusive relationship between svPDE and ENPP3. Interestingly, besides the venom gland, prairie rattlesnakes show a predominant expression of svPDE in the kidney (*Figure 2D*).

## The structure of svPDE and its binding targets

Even though the presence of phosphodiesterase enzymatic activities against ATP, ADP, and/or DNA have been demonstrated in many crude venoms, the exact structural element responsible for such activities is still unclear. We therefore purified the svPDE directly from the crude venom of Taiwan cobra (*N. atra*) via chromatography (*Figure 3—figure supplement 1*) and determined its 3D structure with X-ray crystallography. The structures of unliganded svPDE (apo form, PDB accession: 5GZ4) and liganded svPDE (AMP-complexed form, PDB accession: 5GZ5) were determined at resolutions of 2.55 and 2.09 Å (*Supplementary file 2A*), respectively. Crystal structures show that svPDE resembles human ENPP3 and ENPP1 proteins (the structural homologs with currently available structural data) with *N*-terminal somatomedin domains (SMB1 and SMB2), a catalytic phosphodiesterase domain (PDE) and a C-terminal nuclease domain (NUC) (*Figure 3A and B*). An insertion loop (IL) and two catalytical zinc ions, essential for the ATP catalytical activity, constitute the bimetallic active site of the catalytic PDE domain (*Figure 3C*), and the residues coordinated with zinc ions are highly conserved in svPDE, ENPP3, and ENPP1, indicating a similar nucleobase recognition mechanism. AMP is accommodated in a nucleotide-binding pocket formed by residues W251/E255/Y269 from the IL loop and residues F186/K271 from the catalytic PDE domain (*Figure 3C and D*), where T185 is the nucleophile residue responsible for the enzymatic activity. The conformation of the active site in the apo form is almost identical to that in the AMP-complexed form. When AMP is bound to svPDE, only side chains of N206 and K271 shift and interact with the O atom of the phosphate group and the N6 atom of AMP, respectively (*Figure 3C*). Moreover, the structural overlay of AMP-complexed svPDE, ENPP3, and ENPP1 shows that the lysine claw composed of K255, K278, and K528 in ENPP1, which interacts with the terminal phosphate of ATP, appears differently in svPDE or ENPP3 (*Figure 3—figure supplement 2*). While svPDE contains two conserved lysines (K184 and K455), only K185 is located close to the substrate. This suggests that svPDE has an ENPP3-like substrate specificity due to ENPP3 contains only one conserved lysine (K205) in the active region.

Biochemical measurements of the substrate specificity revealed that svPDE could hydrolyze ATP, ADP, and NAD (nicotinamide adenine dinucleotide) with relatively low $K_m$ compared to the hydrolysis of GTP, CTP, and UTP (*Supplementary file 2B*), indicating that svPDE is a more effective catalyst for adenine-containing nucleotides. Given that the 3D structure and substrates are conserved for both svPDE and ENPP3, svPDE could hydrolyze extracellular nucleotides, showing the preference for adenine nucleotides and derivatives. Partial electron densities of N-glycans show that seven residues of svPDE have been post-translationally modified (*Figure 3E* and *Figure 3—figure supplement 3*). N512, one of the seven N-glycans, and C408-C795 disulfide linkage are essential for the stabilization of the PDE-NUC interface of ENPP members and were found to be conserved in svPDE.

Apart from the enzymatic activities, membrane-attached human ENPP1 has been shown to be able to inhibit insulin receptor (IR) signaling processes (*Kato et al., 2012*), and membrane-attached human ENPP3 plays a functional role in the Golgi apparatus of neuronal cells to suppress the activity of 1,6-N-acetylglucosaminyltransferase, GnT-IX (*Korekane et al., 2013*), and perturbed N-glycosylation functions. Interestingly, mass spectrometric analysis of the N-glycosylation patterns with less terminal sialic acid contrasts sharply with our previous study on venom glycoproteins such as SVMP (*Huang et al., 2015*). Without terminal sialic acids, the exposed lactose disaccharide domain of the N-glycan moiety in svPDE enables its binding to immunologically important galectins. As shown in *Figures 3F and G*, svPDE indeed interacts with IR ectodomain and Gal-3 with apparent binding affinities $K_D$ ~ 1.8 μM and ~269 μM, respectively. A recent study reported that direct binding of Gal-3 to insulin receptors triggers the disease state of mice adipocytes (*Li et al., 2016*). Hence, it would be interesting to investigate how svPDE might interact with IR and Gal-3 and get involved in the tissue damage and enhancement of cell toxicity resulting from VSTs, such as 3FTx and cytotoxins, in cobra snakebites.

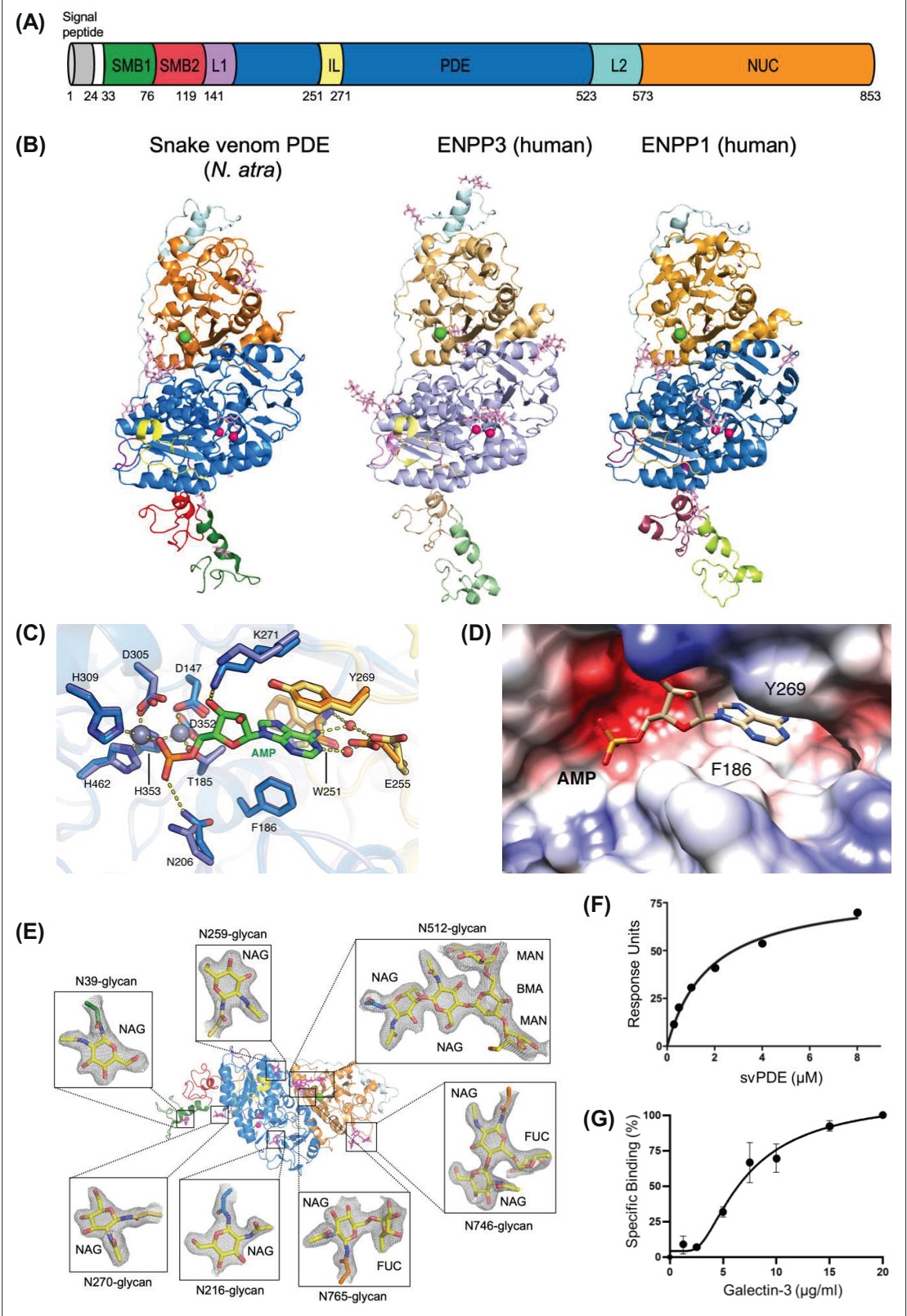

**Figure 3.** SvPDE sharing similar structural folding and binding partners with human ENPP1 and ENPP3.
 (**A**) Domain organizations of *Naja atra* snake venom phosphodiesterase (svPDE). SMB1, somatomedin-B-like 1 domain; SMB2, somatomedin-B-like 2 domain; L1, loop 1; PDE, catalytic phosphodiesterase domain; IL, insertion loop; L2, loop 2; NUC, nuclease-like domain. (**B**) Crystal structures of svPDE from *N. atra* (PDB code: 5GZ4), human ENPP3 (PDB code: 6C01), and human ENPP1 (PDB code: 6WET) in cartoon representation. Zinc and calcium

*Figure 3 continued on next page*

*Figure 3 continued*

ions are shown as hot pink and green spheres, respectively. N-glycans are shown as light pink sticks. (**C**) Superposition of active sites of svPDE from *N. atra* in the apo and AMP-complexed forms. Zinc atoms are shown as gray spheres and water atoms are shown as red ones. Residues involving AMP binding in the apo and AMP-complexed forms are shown in slate/orange and marine/yellow, respectively. Hydrogen bonds and coordinate bonds are shown as dashed yellow lines. (**D**) Electrostatic potential surface of the nucleotide-binding pocket of svPDE from *N. atra* in the AMP-complexed form. (**E**) N-glycans at Asn39, Asn216, Asn259, Asn270, Asn512, and Asn746 of svPDE from *N. atra* are shown as sticks with electron densities. 2Fo-Fc electron density maps contoured at 1.0σ. (**F**) Surface plasmon resonance (SPR) investigation of the binding between svPDE (0.25, 0.5, 1.0, 2.0, 4.0, and 8.0 μM) and the immobilized insulin receptor. $K_D$ of svPDE binding to the insulin receptor were obtained by steady-state affinity model. (**G**) Binding affinity measurement of Gal-3 with svPDE from *N. atra*. Standard deviations of three replicates were indicated.

The online version of this article includes the following source data and figure supplement(s) for figure 3:

**Source data 1.** The original uncropped image of SDS-PAGE.

**Figure supplement 1.** Purification of snake venom phosphodiesterase (svPDE) from the crude venom of *Naja atra* habitated in Taiwan.

**Figure supplement 2.** The structural overlay of AMP-complexed snake venom phosphodiesterase (svPDE), ENPP3, and ENPP1.

**Figure supplement 3.** Identification of N-glycosylated sites and N-glycan patterns of snake venom phosphodiesterase (svPDE) from *Naja atra* based on electron density distributions from X-ray diffraction data and mass spectrometric methods.

**Figure supplement 4.** Putty (sausage) representation of the crystal structures of snake venom phosphodiesterase (svPDE) from *Naja atra*.

**Figure supplement 5.** Nucleotide and NAD-hydrolysis activities of snake venom phosphodiesterase (svPDE) from *Naja atra*.

## Discussion

Since ENPP3 is shared among Metazoan species, but svPDE is limited to snakes (***Figure 1—figure supplement 2***), our result suggests that svPDE has evolved from an ancestral ENPP3 gene by co-option, an evolutionary strategy of using preexisting proteins for new functions. This type of co-option was also used for other minor venom components of 5′ nucleotidase (5NT), as shown in a recent study on the *Bothrops jararaca* genome (***Almeida et al., 2021***). By using an alternative 5′ exon, ENPP3 replaced its domains for membrane attachment by the signal peptide and became a secretory svPDE protein. The mechanism of svPDE expression has changed multiple times during snake evolution, and the one that emerged in the common ancestor of Elapidae and Hydrophiidae was associated with an insertion of mobile elements (***Figure 1C***). New genes evolving from the insertion of mobile elements have been found in various species, such as the genes that arose through L1 and Alu insertion in primates (***Long, 2001***). Based on the transcriptomic data, this mobile element is still active and the genomic changes caused by its insertion may still be happening in the extant species. Such expansion of mobile elements has been observed and proposed to contribute to the adaptation of Hydrophiidae recently (***Peng et al., 2020***).

Notably, the replacement of regulatory elements that evolved multiple times during snake evolution strongly suggests the importance of keeping or improving the expression of svPDE in the mechanism of venom action. Nevertheless, different replacements of exons and regulatory elements that emerged in Viperidae and Elapidae clades could support and be probably associated with the idea that the fangs of these two clades have evolved independently (***Westeen et al., 2020***).

It should be noted that seven ENPP family members ***Borza et al., 2022*** have merged and duplicated into eight copies (two ENPP7 copies) in the ancestor of Gnathostomata, that is, before the divergence of Acanthodians (cartilaginous fishes) and Euteleostomi (the common ancestor of other bony vertebrates), and then followed by a loss of ENPP5 in the snake lineage (***Figure 1—figure supplement 2***). Since snake genomes contain seven ENPP copies (except for the loss of ENPP5) sharing the conserved PDE domain, all copies may be reported as the targets while using svPDE peptide as a query against the genomes and thus may cause the overestimation of svPDE gene copies in other studies (***Rao et al., 2022***).

Comparative transcriptomics is a powerful tool to reveal species-specific or tissue-specific novel transcripts, providing new insights for further studies. For example, the svPDE expression of Lamprophiidae, even higher than several species of Elapidae, indicates the worth of further study for the less-known Lamprophiidae clade to fill the knowledge gap. In addition, although the svPDE transcripts are rare in the glands of Colubridae, Dipsadidae, and Hydrophiidae, the junction spanning reads and/or de novo assembled transcripts indicate their existence in such clades. Interestingly, the svPDE expression in Duvernoy's glands of Colubridae, although low, several species within the

diverse Colubridae clade have been shown to be moderately venomous. The expression of svPDE in the Duvernoy's glands also highlights its potential function despite that Duvernoy's glands exhibit morphological difference from the venom glands of front-fanged snakes. Similar questions are also worthy to be addressed for the rictal glands of pythons, which are believed to be a relic of the secretion system used by ancestral snakes. Furthermore, the predominance of svPDE in prairie rattlesnakes' kidneys (*Figure 2D*) was only found in males, suggesting an unknown physiological role of svPDE in the kidneys of male vipers. It would be interesting to further explore this sexual bias when more data becomes available from different species and individuals.

Based on our biochemical analysis, similar to ENPP3, svPDE can hydrolyze a variety of substrates, including nucleotides and nucleotide derivatives that can induce multiple cellular effects on the venom-exposed tissues. ENPP3 has been reported to downregulate extracellular ATP secreted from basophils and mast cells and to suppress allergic inflammation (*Tsai et al., 2015*), raising a question of what function has adopted from membrane-attached ENPP3 to secretory svPDE. In the cases of venomous snakebites, extracellular ATP released by damaged cells can activate complex physiological responses, such as platelet aggregation, mast cell secretion, inflammation, membrane permeability, vascular function and neurotransmission (*Gordon, 1986*), and can associate with purinergic receptors as a danger signal to initiate immune responses. Another nucleotide metabolite, NAD, is also believed to activate purinergic receptors to alert the immune system if released to the extracellular matrix (*Audrito et al., 2021*; *Haag et al., 2007*). Together, it is conceivable that the removal of extracellular ATP and NAD by svPDE alters the purinergic signaling of cells around the bitten tissues and redirects the immune defense mechanism of the prey in responding to venom actions. Regarding the role of svPDE in the environment around damaged cells as a result of cytotoxic venom toxins, it is important to note that svPDE is a zinc-dependent enzyme that requires an intact bimetallic active site for its catalytic activity. The extracellular concentration of zinc ions is normally high and that zinc homeostasis plays an important physiological function (*Hara et al., 2017*). It is feasible that svPDE carries out its enzymatic activity in the extracellular environment around damaged cells, where zinc concentration may be elevated due to the release of zinc from damaged cells.

Despite the catalytic functions of svPDE, we also demonstrated that svPDE is capable of binding to the IR and Gal-3, suggesting a novel physiological role for svPDE. Previous studies have revealed that ENPP1 directly interacts with IR and inhibits insulin signaling (*Kato et al., 2012*; *Maddux and Goldfine, 2000*), and the binding of Gal-3 to IR causes insulin resistance (*Li et al., 2016*). In similar fashion, the binding of svPDE to the insulin receptor and Gal-3 may result in the perturbation of insulin signaling. Although snakebite symptoms ascribed to the insulin insensitivity and impaired glucose metabolism need to be further investigated, the versatile role of svPDE in cellular metabolism and immune modulation draws attention to the different aspects of venom action.

The co-opted secretory svPDE in snake venom likely evolved to reuse the original intracellular functions of ENPP3 at the extracellular matrix of snakebite wounds. We speculated that, around the damaged tissues, svPDE interferes with the normal cellular physiology in which ENPP family are involved. Although the exact functions and physiological significance of svPDE are still unclear, it reveals that even as a minor venom component, it may play a much more complicated role in toxicity by perturbing signal transduction that was neglected before. In the light of the ubiquitous presence of svPDE in most, if not all, snake venoms, the highly conserved svPDE is a candidate target in developing a generic antidote for venomous snakebites across different species clades. In principle, the current preclinical antidote and repurposed metal chelators for snakebite victims (*Dennis et al., 2020*) could also function via the inhibition of svPDE by chelating zinc ions. In addition, some of the inhibitors for ENPP family enzymes show promise as potential therapeutics for snakebite management. In contrast to the current emergency room treatment for snakebites that relies only on anti-snake venom serums, the generic drug would significantly reduce the burden of snakebites on human health.

## Materials and methods
### Whole-genome assembly of *N. atra* distributed in Taiwan

DNA was extracted from the muscle tissue of a male *N. atra* by using AllPure Genomic DNA Extraction Kit (AllBio Inc, Taiwan). Third-generation sequencing was obtained by using the circular consensus sequencing (CCS) mode on the PacBio Sequel II System resulting in 4563622 HiFi pass reads (56.8 Gb

with an average length of 12.4 kb). Next-generation sequencing was obtained in PE150 on the Illumina NovaSeq 6000 System and preprocessed by fastp (version 0.19.4) (*Chen et al., 2018*) to discard reads with ambiguous N and trim five nucleotides on both ends of the read, resulting in 559566268 clean paired-reads (~78 Gb). PacBio HiFi filtered reads were used for genome assembly by hifiasm (version 0.13-r308) (*Cheng et al., 2021*), resulting in a draft genome in a size of 1.88 Gb and N50 of 29.7 Mb. Assembled contigs were then polished by Nextpolish (version 1.3.1) (*Hu et al., 2020*) with Illumina clean paired-reads. See *Supplementary file 3A–C* for the detailed statistics of PacBio CCS reads, Illumina paired-end reads and the draft assembly, respectively.

## Revealing the ENPP3 genomic exons

The ENPP3 exons were identified in the species with available genomes by an iterative search strategy (*Pan and Lin, 2020*) with HMMER (version 3.2.1) (*Wheeler and Eddy, 2013*). Once the hit was found in a species, its sequence was used to update the HMM profile; as the sequences from more species were involved in a profile, it could more sensitively identify the homologs in distant species. The identified exons were scrutinized for the splicing junctions and cross-checked by aligning with de novo assembled transcripts.

## Identification of svPDE and ENPP3 transcripts

Publicly available RNA-seq reads were processed for quality and adapter trimming by using fastp (version 0.21.0) (*Chen et al., 2018*) and then de novo assembled to reconstruct the transcriptome using Trinity (version 2.13.2) (*Grabherr et al., 2011*). The potential svPDE transcript was selected by removing the contigs that well-aligned to the concatenated joined sequence of conserved ENPP3 exons. Fragmented contigs aligned as substrings were excluded by the consideration that they are assembled with sequencing error reads resulting in failure of extension. Since the precise position of the transcription start site is undeterminable due to the nature of poly-A based library preparation, the longest transcript at the 5'end was used for positioning the genomic upstream region of promoter prediction.

## Prediction of promoters and mobile elements

Transmembrane domain and signal peptide were predicted by TMHMM (version 2.0) (*Chen et al., 2003*) and SignalP (version 5.0) (*Almagro Armenteros et al., 2019*), respectively. The genomic upstream of the transcripts was fetched for searching promoters. The core promoter was predicted by ElemeNT (*Sloutskin et al., 2015*) (version 2). The mobile element DR0148352 was recognized by the exceeding multiple hits of such region in the genomes during the iterative search strategy for exons and annotated by searching the Dfam database (release 3.5) (*Storer et al., 2021*).

## Estimation of svPDE and ENPP3 expressions

The counting of E1-E2 junction spanning reads was conducted instead of the typical mapping approach because ENPP3 and svPDE transcripts share long identical sequences (24 exons), from which the original transcripts of reads cannot be distinguished. Meanwhile, the imprecise end of 5' UTR during sequencing might influence the unfair mapping of the first exon across samples. Thus, the expression level of svPDE and ENPP3 transcripts was estimated by directly counting the reads spanning at least 30 bp (i.e., 15 bp in each exon) at the exon'exon junctions between specific E1 (ENPP3-E1, svPDE-E1a, and svPDE-E1b) and shared E2. In order to avoid the estimation biases caused by different studies, experiments, and individuals, the estimation was calculated by the log2 ratio of read counts for svPDE over ENPP3 transcripts within a sample (i.e., a fold change of svPDE over ENPP3 expression). A count of zero was adjusted to a pseudo-count of 1 to prevent the undefinable log transformation of zero or infinitive values.

## Purification of svPDE from the crude venom of *N. atra*

One gram of crude venom was dissolved in 3 mL of 50 mM phosphate buffer (pH 6.2) with 500 mM NaCl, filtered and applied to a Sephadex G-75 column (General Electric Company), and pre-equilibrated with 50 mM phosphate buffer (pH 6.2). The fraction containing svPDE (the peak highlighted in red in *Figure 3—figure supplement 1A*) was collected and applied to a MonoQ column (General Electric Company). The flow-through fraction (the peak highlighted in red in *Figure 3—figure supplement*

*1B*) was concentrated, dialyzed against 50 mM phosphate buffer (pH 7.4), and applied to a HiTrap Heparin column (General Electric Company). The bound proteins were eluted with 50 mM phosphate buffer (pH 7.4) with a gradient of NaCl rising from 0 to 1 M. The first eluted fraction (the peak highlighted in red *Figure 3—figure supplement 1C*) was concentrated, dialyzed against 35 mM phosphate buffer (pH 7.2), and applied to a MonoS column (General Electric Company). Subsequently, 35 mM phosphate buffer (pH 7.2) with a gradient of NaCl from 0 to 1 M was used to elute the bound proteins. The fraction following the flow-through fraction (the peak highlighted in red in *Figure 3—figure supplement 1D*) was concentrated and subjected to a Superdex 200 column (General Electric Company), pre-equilibrated in 20 mM HEPES (pH 7.4) with 100 mM NaCl. Molecular weight and the purity of svPDE were analyzed with a reducing 10% (w/v) SDS/PAGE followed by staining with Coomassie blue (inset in *Figure 3—figure supplement 5E*).

## Protein crystallization
Crystallization of svPDE was performed at 293 K by the hanging drop vapor-diffusion method. Pure protein solution (12 mg/mL) was mixed with a well solution containing 0.1 M imidazole (pH 6.6), 0.2 M zinc acetate, and 23% PEG 3350 in 1:1 volume ratio. Single crystals were grown to ~0.1 mm in the longest dimension after 15 d. To obtain crystals of svPDE in complex with AMP, the svPDE crystal, grown by mixing 1 μL of protein solution (12 mg/mL) with 1 μL of well solution (0.1 M imidazole pH 6.5, 0.2 M zinc acetate, and 20% PEG 3000), was soaked with 1 mM AMP for 2 d. SvPDE crystals of apo form and AMP-complexed form were cryoprotected by a brief transfer to their respective reservoir solutions supplemented with 20% ethylene glycol before data collection.

## Structure determination and refinement
The data sets of svPDE in the apo form (unliganded) and the AMP-complexed form were collected at BL13C1 and BL15A1 of NSRRC (National Synchrotron Radiation Research Center, Taiwan), respectively. Both data sets were indexed, integrated, and scaled with HKL2000. The structure was solved by molecular replacement with the program Phaser (*McCoy et al., 2007*) using the PDE and NUC domains from mouse ENPP1 (PDB accession: 4B56) as the search model. SMB1 and SMB2 domains from mouse ENPP2 (PDB accession: 3NKM) were used as templates to build the N-terminal structure. Subsequently, structural refinements were performed using Coot (*Emsley et al., 2010*) and PHENIX (*Liebschner et al., 2019*). Crystallographic and refinement statistics are listed in *Supplementary file 2A*. All structure figures were obtained using PyMOL (version 1.3, Schrödinger, LLC) and Chimera (*Pettersen et al., 2004*). The structure in the AMP-complexed form was solved by using the apo form of svPDE as a search model. However, there is no clear electron density map allowing us to build the SMB1 domain of svPDE in the AMP-complexed form, indicating the SMB1 domain could be highly flexible. B factor representations of svPDE in apo and AMP-complexed forms identified the N-terminus including SMB1 and SMB2 as a highly flexible region (*Figure 3—figure supplement 4*).

## Nucleotide and NAD degradation assay
The nucleotide and NAD degradation activities of svPDE from *N. atra* were analyzed by an HPLC equipped with an analytical C18 column (*Figure 3—figure supplement 5*). The reaction mixture containing 40 nM (for nucleotide degradation assay) or 80 nM (for NAD degradation assay) svPDE and 1 mM ATP or ADP or NAD in 20 mM Tris-HCl (pH 8.0) was incubated at 37°C. After various time periods, reactions were terminated by adding 1.0 N NaOH and analyzed by HPLC. In addition, enzyme kinetics of svPDE for nucleotides and NAD were also determined using HPLC. 40 nM (for nucleotide degradation) or 80 nM (for NAD degradation) svPDE was incubated with various concentrations of ATP or ADP or NAD in 20 mM Tris-HCl (pH 8.0) at 37°C. The reactions were terminated by addition of 1.0 N NaOH and analyzed by HPLC. $K_m$ and $V_{max}$ were estimated from a Michaelis–Menten plot of the initial rates as a function of substrate concentrations. $k$cat was obtained using the equation: $k$cat = $V_{max}$ / [E].

## Mass spectrometric analysis
Trypsin digestion of svPDE from *N. atra* was performed at an enzyme to substrate ratio of 1:50 (w/w) after the cysteine residues of svPDE were reduced and alkylated with dithiothreitol and iodoacetamide, respectively. Glycopeptides were enriched by hydrophilic interaction (HILIC) liquid chromatography

using Superdex 75 prep grade resins. The binding and wash buffers for HILIC were 75% acetonitrile (ACN) and 0.1% formic acid (FA). Glycopeptides were eluted from the resins with 50% and then 25% ACN in the presence of 0.1% FA. The dried glycopeptide mixtures were reconstituted with $H_2O$, 0.1% FA, and analyzed by liquid chromatography-tandem mass spectrometry (LC-MS/MS). The glycopeptides were separated on a reverse-phase column (BEH C18, 0.1 × 100 mm, Waters Cooperation) at a flow rate of 300 nL/min using a 60 min ACN gradient in the presence of 0.1% FA. The effluents were analyzed online by an Orbitrap Elite hybrid mass spectrometer (Thermo Fisher Scientific Inc). The mass spectrometer was operated in positive ion mode, and the tandem mass spectra (MS/MS) were acquired in a data-dependent manner. Briefly, the six most intense ions in the survey MS spectrum were selected for collision-induced dissociation (CID) and then scanned in the orbitrap analyzer for obtaining the MS/MS spectra. The collected MS/MS spectra were subjected to glycopeptide identification using the Byonic software (Protein Matric, inc). The database search was performed against a focused database containing four venom glycoproteins (svPDE, L-amino acid oxidase, 5′-nucleotidase, and cobra venom factor) with the following parameters: enzyme specificity, fully tryptic; maximum number of missed cleavages, 2; fixed modifications, carbamidomethylation of cysteine; variable modifications, oxidation of methionine, acetylation of protein N-terminus, and glutamine to pyroglutamate conversion at peptide N-terminus; N-glycans, 309 mammalian N-glycans provided by the Byonic software; mass tolerances, 15 ppm and 100 ppm for MS and MS/MS spectra, respectively. Finally, the matched proteins were filtered to 1% false discovery rates (FDRs) estimated by a decoy database search.

### Solid-phase binding assay

The flat-bottomed, 394-well microliter plate was coated with 50 µg/mL svPDE in the presence of coating buffer (0.1 M $NaHCO_3$, pH 9.4). After 3 hr of incubation at 37°C, the free surface of the wells was blocked by adding 5% BSA/PBS and the plate was incubated at 37°C for an additional 2 hr. Human recombinant Galectin 3 (His tag) (Sino Biological Inc) in PBS was then applied to the wells coated with svPDE after the wells were washed with PBST three times. The soluble Gal3-His was then allowed to interact with immobilized svPDE at 37°C for 16 hr. In order to remove the unbound Gal3-His after reaction, the wells underwent three PBST washes. A mouse 6X His tag antibody (HRP) (GeneTex Inc) was then used to recognize Gal3-His at room temperature for 1 hr. After washing with PBST for three times to eliminate nonspecific interactions, bound Gal3-His was quantified by measuring the absorbance at 450 nm after adding the substrate, 3,3′,5,5′-tetramethylbenzidine (TMB) for horseradish and 1 N $H_2SO_4$ for termination of the reaction.

### Acknowledgements

This research was funded by Higher Education SPROUT Project launched by the Ministry of Education, Taiwan (ROC). CTP and HHC were supported by Yushan Scholar Program. We thank the experimental facility and the technical services provided by the 'Synchrotron Radiation Protein Crystallography Facility of the National Core Facility Program for Biotechnology, Ministry of Science and Technology' and the 'National Synchrotron Radiation Research Center,' a national user facility supported by the Ministry of Science and Technology, Taiwan (ROC). We are also immensely grateful to Gerry Tonkin-Hill and Jessica K Calland for improving the manuscript.

## Additional information

### Funding

| Funder | Grant reference number | Author |
|---|---|---|
| Ministry of Education | Higher Education SPROUT Project | I-Jin Lin |
| Ministry of Education | Yushan Scholar Program | Cheng-Tsung Pan |

The funders had no role in study design, data collection and interpretation, or the decision to submit the work for publication.

## Author contributions
Cheng-Tsung Pan, Conceptualization, Formal analysis, Investigation, Visualization, Writing – original draft, Writing – review and editing, Conceived and planned the study; Chien-Chu Lin, Investigation, Visualization, Writing – original draft, Writing – review and editing, Solved the X-ray crystal structures and performed the biochemical binding experiments; I-Jin Lin, Investigation, Visualization, Writing – original draft, Writing – review and editing, Performed the biochemical binding experiments; Kun-Yi Chien, Investigation, Carried out the experiments of MASS for protein and glycan ID; Yeong-Shin Lin, Advised the bioinformatics analysis and providedcomments; Hsiao-Han Chang, Supervision, Funding acquisition, Writing – review and editing; Wen-Guey Wu, Resources, Supervision, Funding acquisition, Writing – review and editing, Guided the snake venom project and toxin protein group

## Author ORCIDs
Cheng-Tsung Pan http://orcid.org/0000-0002-2820-3067
Chien-Chu Lin http://orcid.org/0000-0002-0647-427X
Yeong-Shin Lin http://orcid.org/0000-0001-9229-0102
Hsiao-Han Chang http://orcid.org/0000-0001-8016-1530

## Ethics
One male Taiwan cobra, raised by Tainan World Snake King Educational Farm, was donated by Tainan County Government and the permit was granted by Hsinchu City Animal Protection and Health Inspection office for this study. The animal use protocol was approved by the Institutional Animal Care and Use Committee of National Tsing Hua University (IACUC Protocol No. 11003H010 and Approval No. 110010).

## Decision letter and Author response
Decision letter https://doi.org/10.7554/eLife.83966.sa1
Author response https://doi.org/10.7554/eLife.83966.sa2

## Additional files

### Supplementary files
• Supplementary file 1. Summary of genomic and transcriptomic data used in this study.

• Supplementary file 2. The summaries of structure and biochemical analyses. (A) Crystallographic and refinement statistics. (B) Direct biochemical measurement of the substrate specificity.

• Supplementary file 3. The summaries of the sequencing and assembly results. (A) Statistics of PacBio CCS reads. (B) Statistics of Illumina paired-end reads (C) Statistics of draft assembly.

• MDAR checklist

### Data availability
The CDS and peptides of N. atra svPDE revealed in this study have been presented in Figure 1-figure supplement 1. The signal peptides of svPDE in different snake species identified with the public genome assemblies have been shown in Figure 1-figure supplement 3. The newly identified first exon for each species could be retrieved by searching its corresponding genome assembly (NCBI accessions in Supplementary File 1).

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
