## [Editor Report]

This manuscript reports important findings regarding the evolution of snake venom proteins. The conclusions are convincing and are based on appropriate and validated methodologies in line with the current state-of-the-art. The findings will be of interest to biologists and biochemists interested in the evolution of venoms as well as those generally interested in the evolution of molecular novelties.

---

## [Decision Letter]

**Decision letter after peer review:**

Thank you for submitting your article "The evolution and structure of snake venom phosphodiesterase (svPDE) highlight its importance in venom actions" for consideration by *eLife*. Your article has been reviewed by 3 peer reviewers, and the evaluation has been overseen by a Reviewing Editor and Christian Landry as the Senior Editor. The following individual involved in the review of your submission has agreed to reveal their identity: Melisa Benard Valle (Reviewer #2).

Essential revisions:

1) There are some technical questions about the crystal structures and the role of ions that should be clarified in the text.

2) There is missing information regarding the samples.

---

## [Author Response]

Essential revisions:1) There are some technical questions about the crystal structures and the role of ions that should be clarified in the text.

Supplementary File 2A has been updated to include the B factor of AMP.

As also suggested by Reviewer 3, the structural overlay of AMP bound to svPDE, ENPP3 and ENPP1 highlights similarities and differences with other ENPP family members, as shown in Figure 3—figure supplement 2.

(Line 257~262, sentences added)

“Moreover, the structural overlay of AMP-complexed svPDE, ENPP3 and ENPP1 shows that the lysine claw composed of K255, K278 and K528 in ENPP1, which interacts with the terminal phosphate of ATP, appears differently in svPDE or ENPP3 (Figure 3—figure supplement 2). While svPDE contains two conserved lysines (K184 and K455), only K185 is located close to the substrate. This suggests that svPDE has an ENPP3-like substrate specificity, due to ENPP3 contains only one conserved lysine (K205) in the active region.”

Since two zinc ions at the active site are essential for the enzymatic activity, it suggests that metal chelators previously reported as preclinical antidotes for the snakebite management [1] presumably could also exert its therapeutic effect via the inhibition of the svPDE enzymatic activities, in responses to the comments by Reviewer 1.

(Line 390~394, sentences added)

“In principle, the current preclinical antidote and repurposed metal chelators for snakebite victims (Dennis et al., 2020) could also function via the inhibition of svPDE by chelating Zinc ions. In addition, some of the inhibitors for ENPP family enzymes show promise as potential therapeutics for snakebite management.”

2) There is missing information regarding the samples.

This information has been included in the Materials and methods (LINE 401) and a supplementary table (Supplementary File 1). We thank the reviewer for pointing this out and updated the main text so that readers can find the information more easily as follows: (LINE 114~115, sentences updated) “included the draft one of Naja atra sequenced from a muscle tissue (ongoing internal project, see Materials and methods).”

(Line 137~138, sentences updated) “with the corresponding genomes available in the NCBI Assembly database (see Supplementary File 1 for sample details).”